# Quantitative EEG (QEEG) Analysis of Emotional Interaction between Abusers and Victims in Intimate Partner Violence: A Pilot Study

**DOI:** 10.3390/brainsci11050570

**Published:** 2021-04-29

**Authors:** Hee-Wook Weon, Youn-Eon Byun, Hyun-Ja Lim

**Affiliations:** 1Department of Brain & Cognitive Science, Seoul University of Buddhism, Seoul 08559, Korea; hwweon@sub.ac.kr; 2Department of Youth Science, Kyonggi University, Suwon 16227, Korea; shinejx@daum.net; 3Department of Community Health & Epidemiology, University of Saskatchewan, Saskatoon, SK S7N 2Z4, Canada

**Keywords:** quantitative EEG, topographical brain map, domestic violence, intimate partner violence, emotional pattern

## Abstract

Background: The perpetrators of intimate partner violence (IPV) and their victims have different emotional states. Abusers typically have problems associated with low self-esteem, low self-awareness, violence, anger, and communication, whereas victims experience mental distress and physical pain. The emotions surrounding IPV for both abuser and victim are key influences on their behavior and their relationship. Methods: The objective of this pilot study was to examine emotional and psychological interactions between IPV abusers and victims using quantified electroencephalogram (QEEG). Two abuser–victim case couples and one non-abusive control couple were recruited from the Mental Image Recovery Program for domestic violence victims in Seoul, South Korea, from 7–30 June 2017. Data collection and analysis were conducted using BrainMaster and NeuroGuide. The emotional pattern characteristics between abuser and victim were examined and compared to those of the non-abusive couple. Results: Emotional states and reaction patterns were different for the non-abusive and IPV couples. Strong delta, theta, and beta waves in the right frontal and left prefrontal lobes were observed in IPV case subjects. This indicated emotional conflict, anger, and a communication block or impaired communication between abuser and victim. Conclusions: Our study findings suggest brainwave control training via neurofeedback could be a possible therapy in managing emotional and communication problems related to IPV.

## 1. Introduction

Globally, the victims of domestic violence are overwhelmingly women, and women tend to experience more severe forms of violence [1]. This is both a major public health problem and a violation of women’s human rights. Most of this violence is intimate partner violence (IPV). IPV, whether it involves married, cohabitating, or non-cohabitating intimate partners, involves violent behavior by a spouse or partner against the other spouse or partner [2]. The term IPV is often used synonymously with domestic abuse or domestic violence [3]. Intimate partner violence can take a number of forms, including physical, verbal, emotional, sexual, or psychological harm. Examples include physical aggression, sexual coercion, psychological abuse, and economically controlling behaviors [4]. According to the World Health Organization, approximately 30% of women who have been in a relationship report that they have experienced some form of physical and/or sexual violence by their intimate partner in their lifetime [4].

The majority of domestic violence victims maintain their relationship with their abusers even after the occurrence of violent behavior. This is due to various internal and external factors, such as economic dependence, mental stressors, and child-rearing problems [2,3,5,6,7,8]. Because family members’ lives are closely linked in Korea, they repeatedly experience an emotional cycle of conflict and reconciliation with varying intensity. Negative emotions penetrate the context of their lives and they are more familiar with criticism, anger, and hostility than praise or positive feelings [9,10]. These complex emotions are often expressed with violence, leading to the abuser–victim relationship.

In general, victims are trapped in domestic violence situations because of isolation, power and control, cultural acceptance, lack of financial resources, fear, shame, or a need to protect children [5]. As a result of abuse, victims may experience physical disabilities, chronic health problems, and mental illness; have financial difficulties; and encounter problems with creating healthy relationships [11,12]. These factors have the potential to contribute to emotional and psychological problems, such as post-traumatic stress disorder and difficult interpersonal relationships due to social isolation and low self-esteem [10]. When the domestic violence process is neglected or repeated, the victims display evidence of a damaged emotional state, manifested as lethargy, reduced motivation, and sleeping disorders [13]. They may express fear, depression, anxiety, and feel they cannot break the cycle of violence for fear of additional violence [14]. Studies of domestic violence abusers show that low self-esteem, high aggression level, and violent behavior relate to the continuation and frequency of violence [15,16]. Studies of victims of domestic violence have shown that the abuser’s violence can easily lead to learned helplessness in the victim, with the victim more likely to accept the violence and return to the abuser even as the intensity of violence becomes more severe [10,17]. The trauma of domestic violence may present as emotional and somatization disorders, such as anxiety, depression, anger, guilt, suicidal thoughts, headache, and gastrointestinal disorders [10,17,18]. Abusers have been found to have relatively low levels of confidence, self-awareness within their families, and, in particular, poor communication skills [9]. The emotions of the perpetrator and victim of IPV are clearly key influences on their behavior and their relationship.

An electroencephalogram (EEG) is a test that records the electrical activity of the brain. This activity produces energy at varying wavelengths (i.e., frequencies, measured in hertz) on a continuum, with individual wave types corresponding to specific frequency ranges (“bands”). Because human brain waves reflect the state of consciousness, many studies use EEGs to identify their characteristics in relation to specific emotional states. Conventional brain wave studies have been used for diagnosis in cognitive neuroscience. EEG technology and the creation of brain electrical signature profiles have also been used to investigate individuals who commit specific crimes [19]. Recently, the use of EEGs has been extended to psychopharmacology. The development of the so-called Pharmaco-EEG is used to observe and explore how drugs alter brain function and to measure drug-induced changes in the brain [20,21]. Because EEG measurements are able to reflect human psychology, leading to the emergence of the field of EEG psychology, it has been proposed that monitoring the emotions of the abuser and the victim in violent domestic relationships would deepen our understanding of these roles. Valuable insights could be gained by identifying the victim’s day-to-day emotional state while co-existing with the abuser, the dynamics of their interaction, and how the relational situation could be improved using scientific data.

Human brain waves recorded by EEG reflect emotional states, and their correlation with emotional reaction and behavior has been studied [22,23]. Studies have shown brain wave characteristics related to problematic behavior and emotional states [24,25]. Brain wave asymmetry in the parietal and frontal lobes reflects low arousal and may indicate major depression [26,27]. An increase in beta waves has been associated with attention deficit-hyperactivity disorder and dyslexia [26]. Alpha wave asymmetry of the frontal lobe (8–13 Hz for adults and 6–9 Hz for infants) is correlated with mood and emotion [28]. Studies have shown that frontal lobe activity asymmetry is related to motivation and personality traits [29,30,31,32]. Previous studies also showed that the left prefrontal cortex (PFC) is connected to higher level activity and that alpha intensity is inversely related to activation level. The right PFC has been associated with withdrawal motivation [33,34,35,36], whereas the asymmetric EEG activity of the left PFC is related to anger [36,37].

Most of the existing studies on domestic violence have focused on the experience of the abuser, the victim, or their children, and there is value in this approach. However, clinical experience suggests that family members who are trapped in situations of domestic violence have misunderstandings about what they are living, have made excuses for it, and have anxiety, all of which may lead to inconsistency in an individual’s responses. Similarly, when a family member involved in domestic violence is asked to identify the feelings or motives of another member, their response may not be correct. Our approach offers the opportunity to examine participants’ emotional characteristics more objectively via brain wave measurements, and, in particular, to evaluate EEG asymmetry. Thus our study objectives were to measure the brain waves of abuser and victim quantitatively, analyze the expected emotional dynamics between the abuser and the victim as suggested by the EEG results, and offer preliminary objective findings regarding areas of emotional and functional limitations in both abuser and victim that may perpetuate IPV.

## 2. Methods

### 2.1. Study Subjects

This study recruited two abuser–victim couples as cases and one couple in a non-abusive relationship as the control. Exclusion criteria for our study were a history of alcoholism, psychiatric disorders, seizures, other mental conditions that prevented completion of the study, cognitive impairments, hypoxic brain injuries, neoplasia, any psychiatric medications, sight or hearing disorders, and dental implants. We recruited abuser–victim couples where (i) a member of a family had complained of domestic violence for at least 10 years, (ii) the victims and their children had been admitted to shelters for a short or long term, and (iii) several police reports on domestic violence existed. Couples attended the Mental Image Recovery Program for domestic violence victims, held jointly by the Brain Science Research Institute at Seoul University of Buddhism and the Korean Victim Support Organization in Seoul, South Korea, from 7–30 June 2017. A non-abusive control couple was selected during the same period and matched by age and family structure. The victims had all consulted a professional counselor at an organization that works with victims of domestic violence and through which an appointment to visit the Institute of Brain Science for the EEG study was made. Participants received preliminary information regarding the EEG evaluation, and the objectives and study protocol were also explained to them. Written consent regarding the use of personal information was received from each participant.

### 2.2. Study Procedure

All EEG diagnosis and consultations were conducted at the Institute of Brain Science, Seoul University of Buddhism, in South Korea. When a participant arrived at the institute, the EEG procedure was explained. After consenting to participate, the individual was asked to wear a cap molded to follow the standard international 10–20 system of electrode placement (Figure 1) [38]. Electrodes were placed over the frontal pole (Fp), frontal lobe (F), central region (C), parietal lobe (P), temporal lobe (T), and occipital lobe (O) of the brain. Nineteen electrodes were placed on the scalp and two electrodes were attached to the left and right earlobes, meaning a total of 21 electrodes were used. The ten-twenty system assigns numbers to the electrodes to indicate position, including laterality. Even numbers correspond to the right hemisphere, odd numbers to the left hemisphere, and “z” to the midline. EEG data were collected from the 19 standardized positions.

### 2.3. Measurements

At the study visit, subjects were seated comfortably in a room with reduced light and noise. They were required to remove all of their jewelry accessories such as earrings and necklaces.

EEG measurements with eyes closed were taken for 7 to 20 min in a stable/resting state by minimizing physical movement and thereby excluding artifacts [39]. The subjects were seated with their eyes closed for 20 min before the measurements to achieve this stable/resting state. The BrainMaster Discovery biofeedback EEG device (Brainmaster Technologies, Inc., Bedford, OH, USA), which was used in this study, conducts measurements for 12 min with eyes closed. Two minutes of the measurement data was used for this study [40]. It is the recommendation of most QEEG experts that sample sizes of 2 to 5 min of artifact-free EEG be used in a clinical evaluation [41,42]. A resting QEEG measurement is concerned with possible asymmetry between the left and right brains, and is sometimes used to diagnose diseases such as dementia [43]. For statistical analysis and artifact removal, we used the 510(k) FDA certified NeuroGuide software (NeuroGuide, Applied Neuroscience, Inc., St. Petersburg, FL, USA). The NeuroGuide provides a Z-score standardized database, which can be used to compare and analyze brainwaves. It is the most commonly used analytics tool because it is both stable and practical in use [44]. Only 90% of Split Half Reliability and 90% of Test Retest Reliability were used for analysis, meaning that abnormal findings had less than a 10% probability of arising by chance [45,46]. The data sets were recorded at a 256 Hz sampling rate and the amplifier band pass was 1–30 Hz. The fast Fourier transform (FFT) method with normalization was utilized to estimate the powers. For removal of artifacts, artifact-free epochs with a minimal duration of 2.5 s were selected offline by an automatic artifact rejection algorithm augmented by visual inspection by electrophysiological expert editors. All electrode impedances were below 5 kiloohms. All of the subjects of this study showed a normal emotional state in their daily life and their social life was smooth. The emotional and psychological information for a specific frequency domain was analyzed for relative power (i.e., intensity of wave type evaluated as a proportion of the overall amount of brain activity in an individual). NeuroGuide provides a Z-score standardized database after EEG measurement, which can be used for comparison between subjects; this software also provides raw patient data and topographical images of the brain. The frequency bands were calculated as delta (1.0–4.0 Hz), theta (4.0–8.0 Hz), alpha (8.0–12.0 Hz), and beta (12.0–25.0 Hz).

### 2.4. Topographical Brain Mapping

The topographical brain map is a graphical representation of the actual measured voltage (i.e., absolute power) in different areas of the brain. The literature shows that QEEG is highly reliable and reproducible [47,48]. The degree of intensity in each region is shown on a color gradient scale for each frequency band and comprehensively presents an individual’s brain wave activity. This technique is commonly used by researchers [28] and is useful for comparing inter-couple and intra-couple brain activity characteristics.

## 3. Results

A total of six adults participated in this study: two abuser–victim pairs as the case couples (denoted by “B” to “E”) and one non-abusive husband-wife pair as the control couple (denoted by “A”). Neither violent participant had drug or alcohol problems or violent behavior outside of the intimate partner relationship. Both victims and their children had received counseling from experts related to domestic violence. All participants were middle-aged and had children. Characteristics of the study participants are presented in Table 1 and Table 2. The study participants’ emotional state, and by extension their relationships, were examined through quantified EEG values and topographical brain map images. Couple A was considered a normal and valid control couple for comparison with abuser–victim pairs B and C because their EEG measurements were within the normal distribution range (Z-score = ±1.96–±2.00), as shown in Table 3. Additionally, the absolute power values for each frequency band in these control subjects fell into the normal distribution range for all 19 channels (Table 4).

### 3.1. QEEG

The relative intensity of the frequency band for each wave type, and the left–right asymmetry in alpha, beta, and high beta wave ranges for each study participant, are shown in Table 5. Emotional patterns can be examined through the left and right relationship of quantified values in the alpha, beta, and high beta wave ranges. In the beta wave range, emotional abnormalities were suggested when the beta wave intensity of the right hemisphere was higher than that of the left hemisphere; the opposite pattern was sought for the alpha waves. When evaluating alpha waves, emotional abnormalities can be suspected when the intensity difference between right and left hemispheres is more than 20% [49] or when alpha wave activity of the left hemisphere is higher than that of the right. Alpha wave intensity in the occipital lobe area suggests mood swings and depression. In the beta wave frequency range, emotional abnormalities can be suspected when the beta wave intensity of the right hemisphere is higher than that of the left hemisphere, which is the opposite of the alpha wave.

Detailed locations of brainwave abnormalities are presented in Table 5 and Figure 2. In summary, the QEEG analysis revealed that Couple B and C showed abnormalities in the frontal lobe region in the delta, theta, and beta wave ranges, with expected short-term memory decline, limitations in problem solving, a lack of emotional control, poor processing ability, depression, social cognitive issues, and hypersensitive reactions. Individually, B showed high delta wave values in the frontal lobe region and C had high values at the prefrontal lobe region. Couple D and E showed abnormalities in the frontal and prefrontal regions in both delta and theta wave ranges. Moreover, in the alpha wave range, B and C showed abnormalities in the occipital lobe region. In contrast, the control couple showed low attention due to aging in the theta range. EEG values of A-1 were generally in the normal range except for theta waves. A-2′s findings suggest emotional abnormalities in the occipital lobe area of the alpha wave range. The results suggested that A-2 may exhibit emotional swings such as depression, whereas A-1 exhibits normal acceptance and flexibility.

### 3.2. Topographical Brain Map

Table 6 provides details of the brain map results, in addition to the expected effect of these abnormal areas on the couple’s emotional relationship. Topographic images are presented in Figure 2. In the control couple, the graphic generally shows similar brain activity in the husband and wife, although appreciably different graphs were generated within the case couples.

In evaluating the overall relational picture of the specific couples, B and C showed opposite emotional patterns. This implies that the possibility of domestic violence is elevated when C reveals emotions impulsively and B cannot accept them with flexibility and appropriate anger control. Couple D and E had a quite different emotional pattern in their husband–wife relationship, such that they had difficulty communicating emotions to each other. In particular, in the delta, theta, and beta wave ranges, D showed a strong avoidance/withdrawal behavior around the prefrontal region, but E revealed strong approach behavior. This implies that when there is an emotional conflict between the couple, E approaches it by actively revealing feelings, but D rejects or avoids these emotions, potentially by violence. The control couple generally showed similar patterns but had some differences in situational perceptions and displays of emotion. Individually, A-2 may show emotional swings, but A-1’s acceptance, flexibility, and cooperativeness preserves the stable emotional communication between them.

## 4. Discussion

Studies show that the abuser and the victim in domestic violence have different emotional states. The abuser often has problems associated with low self-esteem, low self-awareness, violence, anger, and communication [15]. Most victims of domestic violence are women who often have to deal with the complex situations in which, constrained by economic difficulties and child-rearing problems, they must maintain their family even after violence occurs. The abuser and victim most frequently engage each other using negative emotions, such as blame, hostility, irritation, and anger, than with positive ones. They live in an infinite conflict and reconciliation loop [8,18].

Individual emotional characteristics within each couple were analyzed through the quantifiable brainwave measurement of EEG. Couples’ emotional patterns were suggested using topographical brain maps. In our study, we used QEEG to identify the emotional states that affect behavioral patterns through brain waves. The study measured and analyzed the quantified brain waves to outline the emotional associations of partners who could be involved in violence. In general, quantified EEG values and topographical brain maps highlighted emotional processing and problem-solving inabilities, in addition to attention deficit and short-term memory problems.

The emotional state and the reaction pattern of the husband and wife were significantly different between the non-abusive and IPV couples. Based on the quantified values and topographical brain maps, our study suggests that there were fewer differences in the emotional state in the non-abusive couple. These modalities support reciprocity as being important to having a successful relationship; the flexible emotions expressed by one individual were accepted by the other, and emotional problems brought up by the other individual were accepted by the first. Non-abusive and IPV couples are expected to have similar emotional patterns towards problem recognition and finding a solution.

In contrast, domestic violence couples tend to demonstrate contradictory emotional patterns with opposite extremes, indicating emotional uneasiness between the abuser and victim. High EEG asymmetry results in a lack of attention, such as attention deficit hyperactivity disorder (ADHD), low self-esteem, negative emotion, etc. [26]. Compared to a non-abusive husband–wife couple, abuser–victim couples show abnormal EEG activity, suggesting problems such as emotional control, anger, hypersensitivity, depression, anxiety, fear, and low self-esteem [50]. In the current study, however, there was variation between the abuser–victim couples. For example, in the emotional pattern of abuser–victim Couple B and C, the husband and wife could be described as seeing each other and existing in close proximity but being unable to effectively connect with each other. The relationship might have uni-directional communication, a dynamic potentially caused by violence if one individual is unable to flexibly accept the argument or request of the other during an emotional conflict. However, communication could also follow a ‘same direction but different purposes pattern’ (as with Couple D and E) in which the couple subsequently cannot agree. Couple D and E also showed that one had a high activation of the left brain corresponding with social approach behaviors, whereas the other had a high activation of the right brain associated with socially avoidant behavior or withdrawal [51,52]. This could be explained as a communication block that occurs through avoidance or disregard of one party for the actively presented argument or request of the other. In these situations, violence may result.

We expect brain waves to be changed for a period of time after treatment such as neurofeedback therapy. The literature has shown that QEEG was changed after neurofeedback for patients affected by depression, anxiety disorder, post-traumatic stress disorder (PTSD), traumatic brain injury (TBI), violence and aggressive criminal behavior, etc. [53,54,55,56,57,58,59]. Brown et al. (2019) located problem areas of the brain in QEEG and used neurofeedback therapy to treat IPV survivors who experienced head TBI [59]. Their results showed significant differences in the QEEG data following the completion of neurofeedback, which suggested neurofeedback could mitigate symptoms of probable TBI in IPV survivors.

To our knowledge, there are very few studies using QEEG in IPV research. In our study, various emotional explanations were offered for abnormal QEEG activity among IPV couples. Our findings may play a role as a potential marker for identifying a problem area in emotional states among IPV subjects. Brainwave control training via neurofeedback for IPV abusers and victims could be an effective therapy in managing emotional incontinence, anger control, and communication problems related to IPV.

One of this study’s limitations is its very small sample. Given the sensitive nature of the research area, it was challenging to recruit study subjects. Thus, it is not certain that those who agreed to participate are truly representative of the larger IPV population. Testing could not be repeated to evaluate the sensitivity of the results. However, our study is meaningful in that it was able to make objective EEG-based suggestions regarding possible emotional patterns of both abusers and victims. Another limitation is that the EEG measurements of couples were taken individually, not simultaneously. QEEG hyperscanning might be useful to resolve this issue. An EEG hyperscanning system was not available to us. The relational features of husbands and wives who experienced domestic violence, as described by EEG measurement, suggest the possibility of correcting and improving situations through neurofeedback and brain wave control training. For example, if effective control training is conducted for the brain wave frequencies related to emotion and anger management, the issue of domestic violence may be alleviated or improved. This possibility should be further evaluated by clinical studies in which neurofeedback is applied.

## 5. Conclusions

QEEG analysis can be utilized to reveal emotional patterns, states, and behavior. In this study, we examined the emotional and psychological states of IPV abusers and victims using QEEG. The pilot study found that the emotional state and the reaction pattern of the husband and wife were very different between the non-abusive and IPV couples. Our findings suggest that QEEG measurement can be used to identify emotional or behavioral problems. Thus, based on QEEG patterns, clinicians might suggest brainwave control training via neurofeedback, which is a possible effective therapy for managing emotional incontinence, anger control, and communication problems.

## Figures and Tables

**Figure 1 brainsci-11-00570-f001:**
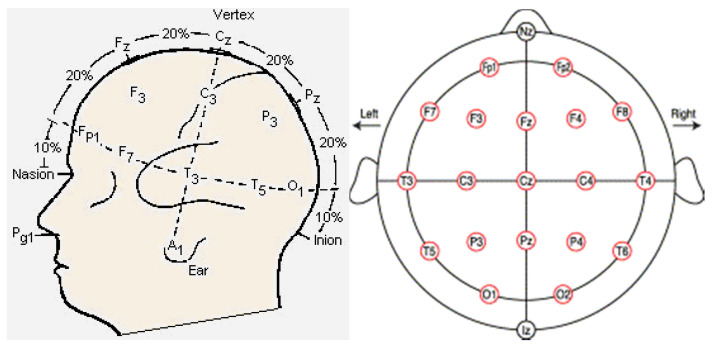
Names and positions of the international 10–20 system used in this study. Nineteen standard positions in the conventional 10–20 system are shown (red circles) [38].

**Figure 2 brainsci-11-00570-f002:**
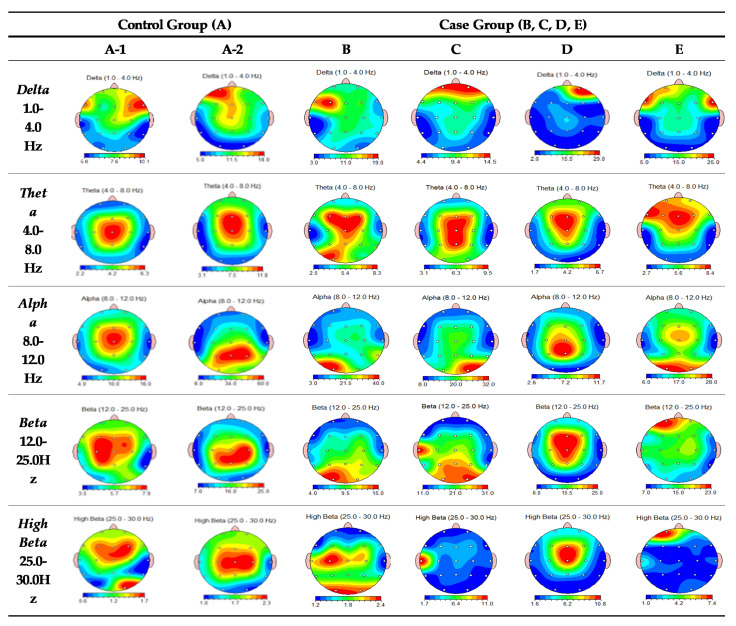
EC FFT absolute power brain mapping.

**Table 1 brainsci-11-00570-t001:** General characteristics of the study participants.

Family Member		Age	Job	Education	No. of Children	Marriage Period (years)
Control	Male(A-1)	51	employee	College	2	20
	Female(A-2)	48	teacher	College		
Case	Male(B)	55	self- employment	Highschool	1	15
	Female(C)	53	housewife	College		
	Male(D)	47	employee	College	3	15
	Female(E)	44	housewife	College		

**Table 2 brainsci-11-00570-t002:** Violence characteristics of the study participants.

Family Member		Period of Violence after the Initial Police Report (years)	Causes of the Use of Violence	Type of Violence
Case	Male(B)	2–3	① Mother–child conflict deepens into husband–wife conflict ② Children’s anger control disorder	Used physical and verbal violence
	Female(C)			
	Male(D)	3–4	① Husband’s economic crisis leads to conflict. ② The wife’s start of economic activities and the burden of mental nurturing	Physical and verbal violence, pressure to use money
	Female(E)			

**Table 3 brainsci-11-00570-t003:** EC FFT absolute power Z-score of the control group (A).

	Control Group (A)									
	A-1					A-2				
	Delta	Theta	Alpha	Beta	High Beta	Delta	Theta	Alpha	Beta	High Beta
**FP1**	−0.36	−1.11	0.14	−0.22	−0.18	0.99	0.47	1.01	0.84	0.19
**FP2**	−0.30	−0.85	0.15	−0.18	−0.18	0.21	0.21	0.86	0.55	0.16
**F3**	−0.56	−0.58	0.38	−0.11	0.13	0.57	0.45	0.98	0.86	0.39
**F4**	−0.11	−0.59	0.34	−0.15	0.26	0.40	0.37	1.12	0.95	0.44
**C3**	−0.34	−0.46	0.30	−0.22	−0.03	0.69	0.34	1.19	1.08	0.77
**C4**	−0.12	−0.51	0.21	−0.41	−0.19	0.48	0.39	1.49	1.36	0.79
**P3**	−0.46	−0.48	−0.15	−0.46	−0.30	0.15	0.08	1.21	0.93	0.85
**P4**	−0.76	−0.70	−0.30	−0.76	−0.60	0.28	0.24	1.28	0.90	0.65
**O1**	−0.54	−0.99	−0.82	−1.18	−0.64	−0.62	−0.55	0.67	−0.07	−0.17
**O2**	−0.50	−0.98	−0.63	−0.72	0.44	−0.70	−0.59	0.51	−0.20	−0.22
**F7**	0.27	−0.54	0.33	0.39	0.44	0.52	0.16	0.81	0.96	0.65
**F8**	0.35	−0.59	0.30	0.12	0.26	0.38	0.23	1.09	1.05	0.77
**T3**	0.06	−0.41	0.17	−0.33	−0.29	0.31	−0.24	0.57	0.25	0.12
**T4**	0.63	−0.22	0.01	−0.57	−0.56	1.21	0.33	0.99	0.48	−0.08
**T5**	−0.07	−0.75	−0.38	−0.69	−0.40	−0.33	−0.55	0.64	0.15	0.33
**T6**	−0.38	−0.90	−0.50	−1.16	−0.60	0.41	0.17	1.08	0.53	0.25
**Fz**	−0.47	−0.67	0.34	−0.18	0.26	0.69	0.49	1.10	1.00	0.69
**Cz**	−0.41	−0.48	0.31	−0.46	−0.06	0.61	0.45	1.34	1.08	0.55
**Pz**	−0.66	−0.50	−0.14	−0.66	−0.50	0.29	0.20	1.21	1.04	0.79

**Table 4 brainsci-11-00570-t004:** Groups’ EC FFT absolute power data result.

	Control Group (A)										Case Group (B, C)										Case Group (D, E)									
	A-1					A-2					B					C					D					E				
	Delta	Theta	Alpha	Beta	High Beta	Delta	Theta	Alpha	Beta	High Beta	Delta	Theta	Alpha	Beta	High Beta	Delta	Theta	Alpha	Beta	High Beta	Delta	Theta	Alpha	Beta	High Beta	Delta	Theta	Alpha	Beta	High Beta
**FP1**	8.08	2.62	7.59	5.67	1.26	17.90	7.29	16.47	11.28	1.72	9.10	4.17	8.97	4.89	1.31	13.85	4.59	10.97	11.94	2.13	10.62	4.38	4.78	11.50	3.99	20.50	7.36	15.13	22.75	7.32
**FP2**	8.61	2.84	8.25	5.86	1.28	12.50	6.88	16.80	10.70	1.79	9.54	4.08	9,74	4.98	1.27	14.42	4.71	11.43	11.84	2.45	28.97	4.48	4.50	11.10	4.12	15.93	6.74	14.20	17.52	3.70
**F3**	7.19	4.59	11.82	7.43	1.58	12.72	9.24	20.86	14.60	1.93	18.55	8.24	12.85	7.64	1.83	8.04	7.51	16.59	18.87	3.56	6.38	5.81	7.39	21.63	6.97	13.57	7.77	18.29	15.09	1.58
**F4**	9.32	4.78	11.79	7.07	1.65	11.93	8.89	24.36	14.83	1.92	10.98	7.62	15.67	8.72	1.88	9.96	7.93	17.09	19.08	3.90	6.72	4.96	5.91	19.37	7.20	11.85	7.57	18.14	15.70	1.66
**C3**	6.98	4.89	12.21	7.88	1.35	12.27	8.35	29.17	19.99	2.11	7.80	5.32	11.83	8.60	2.40	7.17	7.32	15.95	22.11	3.49	5.49	4.51	8.44	19.28	5.34	10.71	6.18	17.59	14.89	1.43
**C4**	8.16	4.81	11.80	6.84	1.22	11.09	8.70	40.71	24.27	2.23	9.10	6.28	17.12	10.52	2.00	7.84	6.78	16.28	20.67	2.86	4.30	3.78	6.72	16.95	5.04	10.23	5.85	18.20	16.13	1.44
**P3**	6.30	4.43	10.10	6.92	1.02	9.20	6.87	50.54	20.02	1.91	7.86	6.12	18.84	11.30	1.77	6.42	7.34	16.71	26.10	2.72	4.48	3.57	8.86	15.82	3.36	10.25	5.15	13.54	13.89	1.47
**P4**	5.86	3.95	9.36	5.78	0.85	10.39	7.85	59.43	20.03	1.74	8.17	5.73	16.89	11.39	1.69	6.44	6.88	18.16	24.91	2.40	3.81	3.13	6.95	12.98	2.93	9.55	4.71	14.92	12.85	1.32
**O1**	5.85	2.95	5.63	4.55	0.84	5.42	4.19	44.24	10.81	1.24	7.46	7.65	39.71	14.57	2.33	6.11	6.78	20.69	27.70	2.33	2.66	2.20	3.35	6.46	1.62	5.86	3.22	27.63	11.42	1.29
**O2**	6.22	2.91	7.29	5.98	1.69	5.43	3.97	36.31	9.08	1.10	5.51	5.40	20.73	12.10	2.28	5.92	6.23	31.57	29.57	2.31	2.62	1.96	3.18	6.59	1.71	6.22	3.20	25.40	10.15	1.81
**F7**	9.37	3.13	7.41	6.38	1.32	10.45	4.82	11.06	9.10	1.54	16.46	5.26	7.12	4.38	1.23	9.51	4.40	9.33	11.67	2.11	5.70	3.15	3.77	9.39	2.64	23.37	8.31	11.86	11.87	1.90
**F8**	10.06	2.92	7.03	5.47	1.25	9.80	4.92	13.55	9.76	1.74	6.13	3.87	10.09	5.76	1.51	7.73	4.27	11.24	12.75	2.27	5.72	2.77	3.03	8.66	2.91	24.96	6.13	10.88	10.33	1.44
**T3**	5.64	2.79	5.89	5.01	0.95	6.41	3.13	8.23	7.55	1.38	3.74	2.54	3.68	6.71	2.08	4.88	3.78	9.86	30.35	10.98	3.23	2.23	3.46	9.38	2.45	5.55	3.08	7.81	15.02	2.94
**T4**	5.92	2.55	4.72	3.70	0.61	7.52	3.74	10.23	7.87	1.13	7.23	3.23	8.61	5.13	1.23	4.84	3.18	8.39	13.15	3.01	5.06	1.72	2.62	6.39	1.95	5.44	2.71	6.46	9.19	1.65
**T5**	6.01	2.80	6.14	4.78	0.82	5.01	3.26	20.02	8.81	1.41	3.86	4.51	18.60	7.76	1.79	4.44	4.24	13.04	16.74	1.87	4.00	2.44	4.08	8.19	1.92	5.85	2.97	12.89	10.69	1.19
**T6**	5.04	2.29	6.09	3.57	0.75	7.90	5.27	39.96	11.75	1.32	7.88	5.37	28.39	10.25	1.68	5.00	3.88	18.31	15.11	1.79	2.94	2.05	4.19	7.86	1.86	5.28	2.95	12.29	7.98	1.04
**Fz**	7.74	5.17	13.63	7.24	1.52	15.15	11.72	28.23	16.77	1.99	10.82	7.61	15.12	7.29	1.63	9.12	8.55	18.11	19.21	3.54	6.31	6.61	8.30	24.94	9.84	14.22	8.38	21.18	15.26	1.42
**Cz**	8.24	6.27	15.39	7.34	1.54	14,69	11.58	40.88	21.31	2.25	10.69	7.92	17.23	8.82	2.11	8.08	9.47	21.15	23.70	3.61	8.54	6.29	10.23	24.72	10.77	13.30	7.37	23.01	16.23	1.47
**Pz**	6.84	5.21	12.14	6.61	0.97	11.83	8.73	59.66	23.32	1.91	9.76	7.00	14.60	11.39	1.73	8.31	9.33	20.35	27.15	2.92	6.46	5.00	11.69	20.33	4.23	11.38	5.52	19.06	13.51	1.40

**Table 5 brainsci-11-00570-t005:** Summary of EEG findings by wave type.

Wave	Feature	Control Group (A)		Case Group (B, C, D, E)			
		Subject A-1	Subject A-2	Subject B	Subject C	Subject D	Subject E
Delta wave	Increased region	F8	Fp1	F3, F7	Fp1, Fp2	Fp2	F7, F8
Theta waves	Increased region	Fz, Cz, Pz	Fz, Cz	F3, F4, Fz, Cz, O1	Fz, Cz, Pz	F3, Fz, Cz	Fp1, F3, F4, F7, Fz, Cz
Alpha waves	Increased region	C3, Fz, Cz, Pz	Pz, P3, P4			Cz, Pz	
	Abnormal region		Asymmetry of O1/O2 *	Asymmetry of O1/O2 ^†^		Asymmetry of F3/F4	Asymmetry of O1/O2 ^++^
Beta and high beta waves	Increased region	F3, F4, Fz, Cz, C3, O2	C3, C4, Cz, Pz	O1, O2, C3	O1, O2, T3, Pz	F3, Fz, Cz, Pz	Fp1
	Abnormal region		Asymmetry of C3/C4	Asymmetry of F3/F4	Asymmetry of F3/F4, F7/F8		

Alpha waves were considered to have increased if asymmetry >20% or left values > right; beta waves were considered to have increased if right > left. * Left vs. right, 44.23 μV^2^ vs. 36.31 μV^2^; ^†^ left vs. right, 39.71 μV^2^ vs. 20.73 μV^2^; ^++^ left vs. right, 27.63 μV^2^ vs. 25.40 μV^2^.

**Table 6 brainsci-11-00570-t006:** Summary of topographical brain map findings by wave type.

Wave	Feature	Control Group (A)		Case Group (B, C, D, E)			
		Subject A-1	Subject A-2	Subject B	Subject C	Subject D	Subject E
Delta waves	Increased region	F8	Fp1	F3, F7	Fp1, Fp2	Fp2	F7, F8
Theta waves	Increased region	Fz, Cz, Pz	Fz, Cz,	F3, F4, Fz, Cz, O1	Fz, Cz, Pz	F3, Fz, Cz	Fp1, F3, F4, F7, Fz, Cz
Alpha waves	Increased region	C3, Fz, Cz, Pz	Pz, P3, P4			Cz, Pz	
	Abnormal region		Asymmetry of O1/O2 *	Asymmetry of O1/O2 ^†^		Asymmetry of F3/F4	Asymmetry of O1/O2 ^++^
Beta and high beta waves	Increased region	F3, F4, Fz, Cz, C3, O2	C3, C4, Cz, Pz	O1, O2, C3	O1, O2, T3, Pz	F3, Fz, Cz, Pz	Fp1
	Abnormal region		Asymmetry of C3/C4	Asymmetry of F3/F4	Asymmetry of F3/F4, F7/F8		

* Left vs. right, 44.23 μV^2^ vs. 36.31 μV^2^; ^†^ left vs. right, 39.71 μV^2^ vs. 20.73 μV^2^; ^++^ left vs. right, 27.63 μV^2^ vs. 25.40 μV^2^.

## Data Availability

The data is not publicly available due to data privacy regulations.

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
