# Peer review of "Quantitative EEG (QEEG) Analysis of Emotional Interaction between Abusers and Victims in Intimate Partner Violence: A Pilot Study"

_brainsci, 2021, doi:10.3390/brainsci11050570_

Round 1

Reviewer 1 Report

I found this  manuscript exciting and its organization excellent. The authors  provided  full details  of the experiments and comprehensive  tables with results.

What to my opinion is missing and I am considering  it  as prerequisite for completion  of this study is the data analysis  with resting EEG hyperscanning  method. E.g. select from  the review paper  ( Quant Imaging Med Surg. 2018 Sep; 8(8): 819–83 doi: 10.21037/qims.2018.09.07   ) the section EEG hyperscanning , further information can be found in Neuroscience Research 2015 Jan;90:25-32.doi: 10.1016/j.neures.2014.11.006.  

Author Response

EEG hyperscanning method should be measured by recruiting research subjects again, but it was practically impossible to call and measure domestic violence perpetrators and victims again. We will use this method in our next study. Instead, a discussion was written to suggest ways to use EEG measurements for domestic violence perpetrators and victims. Please review the manuscript.

Reviewer 2 Report

In this study, authors attempted to identify the emotional status for intimate partner violence (IPV) participants by using qEEG. It is an important contribution to understanding the neurobiological mechanism in dysfunctional behaviour. However, the evidence is not sufficiently strong in supporting the finding rhat there is a link between IPV and emotional status because of several serious drawbacks in the study.

Here is my comments and concerns

# Lack of the screening procedure and quantitative assessment of the violence risk

Authors need to clarify the characteristics of all participants. Authors did not provide any information about the criteria for screening to recruit the participants. For example, did the violent participants have any violence outside of the intimate partner relationship? Did the violent participants have any drug or alcohol issues? In addition, authors should report quantitative assessment of the violence and relationship distress rating of participants.

# Measurement of emotional status

There are many important missing details in the EEG measurements, such as sampling rate, algorithms of power density, and removal of artifacts. In addition, the authors did not justify using only 12 min of EEG signal from 20 min eye closed resting state.

The main problem is, the resting state with short duration is not sufficient to reveal the emotion for the IPV with only two IPV couples and one control. Did authors measure the emotional status during the resting state? Was it is the same emotional status during the violent performance? Because authors did not provide any information of the emotional assessment for the participants, it is difficult to deduce any correlation between the brain wave and emotional rating. These are critical questions that authors need to answer.

# Statistical analysis issues

How did authors define the emotional abnormalities as the definition of 20% difference of spectral power between two hemispheres? Additionally, the ‘Expected emotional/ functional difficulty’ in the Table 4 and 5 do not belong to the result section because they are speculation and interpretation, which should be in the discussion. Moreover, from Table 4 and 5, I cannot see how the authors claim a correlation between the asymmetric findings and emotional/ functional difficulty. For example, why the asymmetric theta waves in Fz, and Cz in control group is related to ‘age related inattention’ and the same finding in the IPV group B is related to ‘short-term memory, attention, depression, and emotional control problem’? Authors need to clarify their claim and provide evidence to support the conclusion.

# Discussion section

Authors need to address if the brain waves changed after the treatment, since the IPV couples attended counselling sessions for a period of time. Overall, it is difficult for me to follow the discussion, authors need to re-write for better flow and clarity of interpretations.

Taking all these issues into account, I cannot be persuaded by these research findings and their presentation.

Round 2

Reviewer 1 Report

If something was wrong in the design of a  study,  it either should  be revised or rejected. The answer of the authors  is not scientifically sound and my critique to the current version follows;  

  1. The design is wrong; if the study is between couples, then the biomarkers should be extracted from the couples not from each subject separately . I do not consider   statistically significant results coming out from a tiny group of subjects.
  2. Accepting the authors’ objection, hard to recruit IPV couples, I suggest to increase the number  of normal couples.
  3. An example of the sloppiness of writing the study is the fact of the arbitrary introduction of feedback as a means of treating domestic violence.  
  4. From a pure static study (resting EEG) the authors are making  comments and recommendations  for stability and unstable conditions of violence between couples.   

Author Response

Response to Reviewer #1 (Round 2)

The study is interesting and the paper is generally well done.  A few items could be improved by a small amount of additional discussion.  Comments:

If something was wrong in the design of a study, it either should be revised or rejected. The answer of the authors is not scientifically sound and my critique to the current version follows;

  1. The design is wrong; if the study is between couples, then the biomarkers should be extracted from the couples not from each subject separately. I do not consider statistically significant results coming out from a tiny group of subjects.

Response: Thank you. If we have EEG hyperscanning, we would have the biomarkers extracted from the couples simultaneously. However, as we mentioned in Round #1,  at the time of this study in 2017, EEG hyperscanning system was not available in Korea and thus, the EEG method was commonly used the brain wave studies.

  1. Accepting the authors’ objection, hard to recruit IPV couples, I suggest to increase the number of normal couples.

Response: Thank you. As you suggest, we wish to increase the number of the normal couples. However, especially under COVID-19 situation in Korea, it is not easy to recruit volunteers from normal couples who were matched to the case couples. Additionally it is practically impossible since our lab on campus was closed and no one is allowed to access it since beginning of the pandemic (March, 2020).

  1. An example of the sloppiness of writing the study is the fact of the arbitrary introduction of feedback as a means of treating domestic violence.

    Response: Thank you. We revised the text to clarify some unclarified sentences. Also we added the following in Discussion:

Literatures already showed that qEEG was changed after neurofeedback for patients with depression, anxiety disorder, PTSD, traumatic brain injury (TBI), violence and aggressive criminal behavior, etc. [54,55,56,57,58,59,60]. The study by Brown et al (2019) located problem areas of the brain in qEEG and used neurofeedback therapy to treat IPV survivors who experienced head TBI[60]. Their results showed significant differences in the qEEG data following the completion of neurofeedback, which suggested neurofeedback could mitigate symptoms of probable TBI in IPV survivors. To our knowledge, there was very few study using qEEG in research on neurofeedback therapy for domestic violent abusers and victims.

In our study, various emotional explanations have been offered for abnormal qEEG activity among IPV. Our findings may play a role as a potential marker or problem area in emotional status among IPV subjects. Brainwave control training via neurofeedback for IPV abusers and victims could be a poss therapy in managing emotional incontinence, anger control, and communication problems related to IPV.

  1. Moore, N. C. (2000). A review of EEG biofeedback treatment of anxiety disorders. Clinical Electroencephalography, 31(1),1-6.

*55. Hammond, D. C. (2005). Neurofeedback treatment of depression and anxiety. Journal of Adult Development, 12(2),131-137.

*56. Thatcher, R. W, North, D., Curtin, R ., et al. (2001). An EEG Severity Index of Traumatic Brain Injury. Neuropsychiatry and Clinical Neuroscience, 13(1),77-87.

* 57.Bounias, M., Laibow, R. E., Bonaly, A. and Stubblebine, A. N. (2001). EEG-neurobiofeedback treatment of patients with brain injury: Part 1: Typological classification of clinical syndromes. Journal of Neurotherapy, 5(4), 23-44.

* 58.Frank Pillmann, Anke Rohde, Simone Ullrich, Steffi Draba, Ursel Sannemüller, and Andreas Marneros. Violence, Criminal Behavior, and the EEG.The Journal of Neuropsychiatry and Clinical Neurosciences 1999 11:4, 454-457

*59. Zukov, I., Ptacek, R. & Fischer, S. EEG Abnormalities in Different Types of Criminal Behavior. Act Nerv Super 50, 110–113 (2008).

* 60.Brown J, Clark D, Pooley AE. Exploring the Use of Neurofeedback Therapy in Mitigating Symptoms of Traumatic Brain Injury in Survivors of Intimate Partner Violence. Journal of Aggression, Maltreatment & Trauma. 2019;28(6):764-783.

  1. From a pure static study (resting EEG) the authors are making comments and recommendations for stability and unstable conditions of violence between couples.  

Response: Thank you. We acknowledge that we would have been able to provide more meaningful and valuable results if we have a larger study samples. However, we would like to emphasize again that this study was based on the Neurofeedback theory of balancing left and right brains through qEEG. 

Reviewer 2 Report

The authors did not address the issues I raised and modify the article properly. A bunch of flaws in the current study:

  1. Definition of IPV: Authors did not clarify the quantitative assessment of IPV.
  2. Experiment design: The experiment design is not appropriate to the study aims.
  3. Statistical analysis: The results based on the asymmetric index and individual topographic maps are not reliable and valid.
  4. Conclusion: with an extremely small sample size and lack of proper statistical analysis, I think the authors over-interpreted the concluding statements.

Author Response

Response to Reviewer #2 (Round 2)

The authors did not address the issues I raised and modify the article properly. A bunch of flaws in the current study:

  1. Definition of IPV: Authors did not clarify the quantitative assessment of IPV.

Response: Thank you and sorry. We added Definition of IPV by WHO (2016) in Introduction: Intimate partner violence can take a number of forms, including physical, verbal, emotional, sexual, or psychological harm, including physical aggression, sexual coercion, psychological abuse, and economically controlling behaviors.”

Unfortunately, our data did not have any quantitative assessment of the violence and relationship distress rating of participants.

  1. Experiment design: The experiment design is not appropriate to the study aims.

      Response: Thank you. We could not locate the term “experiment design” in the text.

Our study is not an experimental study, but is a kind of matched case-control study.

  1. Statistical analysis: The results based on the asymmetric index and individual topographic maps are not reliable and valid.

Response: Thank you. We understand our study sample size too small to claim any “reliability” and/or “validity” of the results. We rather claim our study as an exploratory study or a pilot study for generating hypothesis. We need a large study to confirm our findings.

  1. Conclusion: with an extremely small sample size and lack of proper statistical analysis, I think the authors over-interpreted the concluding statements.

Response: Thank you. We agree with you that our findings is over-interpreted as stated in Conclusion. Before we evaluate this possibility in clinical studies, we definitely need another study with bigger sample size. Thus, we deleted that “This possibility should also be further evaluated by clinical studies applying neurofeedback.”

Our conclusion is:

“We examined the emotional and psychological interactions of IPV abusers and victims using QEEG. The study might suggest that brainwave control training via neurofeedback could be a possible therapy in managing emotional incontinence, anger control, and communication problems related to IPV. To check this possibility, a future study with bigger samples are needed.”

Round 3

Reviewer 1 Report

Ι understand the  difficult  conditions for research and especially to recruit subjects for long physiological studies, but in the publications we can't make discounts.

My opinion remains  the same, this manuscript can't be published under its current form.

Author Response

Response: Thank you. I hope you understand the situation we could not have more recruitment for time being due to COVID-19. After the COVID-19 pandemic is controlled in Korea, we plan to continue this kind of study to confirm our findings.

Reviewer 2 Report

The authors have answered all the questions accurately and responded to all concerns. The paper can be published in the current form.  

Author Response

Thank you very much!